# The Multistage 20-m Shuttle Run Test for Predicting VO_2_Peak in 6–9-Year-Old Children: A Comparison with VO_2_Peak Predictive Equations

**DOI:** 10.3390/biology11091356

**Published:** 2022-09-16

**Authors:** João Paulo Brito, Christophe Domingos, Ana Fátima Pereira, João Moutão, Rafael Oliveira

**Affiliations:** 1Sports Science School of Rio Maior–Polytechnic Institute of Santarém, 2040-413 Rio Maior, Portugal; 2Life Quality Research Centre, 2040-413 Rio Maior, Portugal; 3Research Centre in Sport Sciences, Health Sciences and Human Development, 5001-801 Vila Real, Portugal; 4Polytechnic Institute of Setúbal, Campus do IPS, 2910-761 Setúbal, Portugal

**Keywords:** peak oxygen consumption, regression equations, estimation, shuttle-test, preadolescents

## Abstract

**Simple Summary:**

Cardiorespiratory fitness is one of the main components of physical fitness. For children, a simple test that can be used to assess cardiorespiratory fitness is the multistage 20-m shuttle run test (20mSRT). Research has often used a portable gas analyzer to measure cardiorespiratory fitness in clinical and scientific settings; however, this may not be practical due to the high cost of the device. Moreover, the use of such a device with children is almost impracticable in school environments. Thus, to avoid using such a device, one possibility is to use equations for predicting peak oxygen consumption, which is recognized as one of the best indicators of aerobic fitness. In the present study, 22 equations were used to determine which predictive equations had greater agreement with VO_2_peak values measured by direct oximetry through performance of the 20mSRT. Furthermore, we verified if wearing and carrying a portable gas analyzer constrained the children’s performances. To accomplish these aims, 67 boys and 63 girls were included in the analysis. Our results showed that only six predictive equations correctly predicted the peak oxygen consumption. In addition, for girls, higher values of maximal speed, total laps, and total time were found when a portable gas analyzer was used. This information is helpful to strength and conditioning professionals and to schoolteachers if portable gas analyzers are unavailable or if the environment is not suitable for such assessments.

**Abstract:**

This study aimed (i) to verify if using and carrying a portable gas analyzer (PGA) constrained the performance of school children on the multistage 20-m shuttle run test (20mSRT), (ii) to verify which peak oxygen consumption (VO_2_peak) predictive equations have greater agreement with VO_2_peak values measured by direct oximetry using the 20mSRT. The study participants were 130 children ((67 boys (age 7.76 ± 0.97 years) and 63 girls (age 7.59 ± 0.91 years)), who performed two randomized trials of the 20mSRT with and without a PGA. Twenty-two predictive equations predicted the VO_2_peak values through the performance of the test with and without a PGA. Without a PGA, lower values of maximal speed (MS), total laps (TL), and total time (TT) were found for girls than for boys with a PGA. Only six equations were considered to correctly predict VO_2_peak. In general, higher MS, TL, and TT values were found with the use of a PGA. The predicted VO_2_peak values from the 20mSRT varied significantly among the published predictive equations. Therefore, we suggest that the six equations that presented satisfactory accuracy could be practically used to examine cardiorespiratory fitness in schools and in research with large populations when direct measurement of VO_2_peak is not feasible.

## 1. Introduction

Cardiorespiratory fitness (CRF) is a component of physical fitness commonly used in schools [1,2,3] that can be assessed, in children and adolsecents, through oxygen consumption (VO_2_) during a maximal test [4] to obtain the peak oxygen consumption (VO_2_peak) value, which is usually recognized as one of the best indicators of aerobic fitness [5,6].

In young people, VO_2_peak is reached in the final stage of a maximal effort [7]. Children and adolescents with low CRF tend to present a higher risk of developing metabolic syndrome [8], cardiovascular diseases [9], and depression [10]. In addition, low CRF has strong associations with cardiovascular risk factors such as atherosclerotic vascular disease and abdominal adiposity with origins in pediatric years [11,12], which supports the association between CRF and health-related outcomes in youth and highlights the importance of CRF assessment in children. For instance, low CRF in children has been shown to negatively impact the functional ability of daily tasks and consequently affect life quality. Moreover, it has been reported that low CRF maintenance, from the early years, tends to continue over time [13]. This information highlights the importance of assessing CRF in children to understand their status, since VO_2_peak is considered to be a gold standard assessment [4].

In the school context, indirect field tests are mostly used, as they generally demand low cost, shorter execution time, and are easy to apply with a higher number of participants [14]. The 20-m shuttle run test (20mSRT), designed by Léger et al. [15], is one of the most used field protocols for youth, and it is included in different batteries of CRF tests [5,16]. The 20mSRT requires a small area and almost no equipment, and it can be performed with several participants simultaneously which consequently increases their motivation [17].

To predict VO_2_peak, Léger et al. [15] developed an equation for 8–19-year-old youth and reported a correlation of r = 0.71 between predicted VO_2_peak and a retro-extrapolation VO_2_ measured in the final stage of the test [2]. Recently, a systematic review showed moderate to strong evidence for five equations [1] to predict VO_2_peak values using the 20mSRT in youth [15,18,19,20]. However, despite the results of the previous studies that showed a relationship between VO_2_peak values directly measured and predicted using equations with a range between r = 0.71 and 0.96 [1,2], other studies have tested the validity of the equations and found that they had low performances for predict VO_2_peak [21,22,23,24].

The prediction of VO_2_peak from the 20mSRT does not include variables such as height, weight, body mass index, body surface area, or skinfold measurements which have been recommended to improve the validity of the results [20,25]. Others have suggested that sex and age could affect the predictive power of the 20mSRT [1,18,20,26,27]. However, several predictive equations for esstimating VO_2_peak have presented varied results due to the different characteristics of participants, such as age, maturation, sex, and body composition [26,27]. In this sense, an equation that can effectively predict VO_2_peak should be validated and should produce small variations in amplitude among the predicted results [22].

Despite the proven usefulness of VO_2_peak for achieving quality and consistency of data in children, its use remains to be a challenge [16,28]. The 20mSRT is the most used test in the school context for estimating CRF through the prediction of VO_2_peak values [1,3,16,29], but the validity of predictive equations is still questioned by some authors [2]. Hence, it is relevant to determine which predictive equations provide the best accuracy for children and youth populations, as well as which variables provide more validity to those equations [1].

To measure CRF in field tests conducted in clinical and scientific settings, studies have often used portable gas analyzers (PGA) [29,30,31,32,33]. However, few studies have investigated using the K4*b*^2^ PGA (COSMED, Rome, Italy), one of the frequently used devices [34,35,36], and its influence on the 20mSRT performance. A PGA consists of a face mask attached with a head harness, monitoring wires, and a portable unit fitted with a battery pack in a harness adjusted to a child’s trunks, which weighs approximately ~1 kg. Wearing a PGA can produce discomfort for participants and, consequently, a negative CRF performance [36]. It seems that only Selvadurai et al. [30] and Itaborahy et al. [35] analyzed the interference capacity of a PGA during a maximal stress test in children, although with different procedures.

Given the previous concerns, in this study, we aimed (i) to verify if using and carrying a PGA constrains the performance of children on the 20mSRT, (ii) to verify which of the VO_2_peak predictive equations has greater validity as compared with VO_2_peak values measured by direct oximetry from the 20mSRT, (iii) to analyze if fat mass influences the results of the predictive equations for predicting VO_2_peak values. The following hypotheses were stated: (1) Using a PGA may constrain CRF performance. (2) Some existing VO_2_peak predictive equations are not validated. (3) Fat mass may influence the results of the predictive equations.

## 2. Materials and Methods

A non-probabilistic convenience sampling method was used to recruit participants. Before the study, all children provided verbal consent, and their parents/guardians provided written informed consent. In addition, this study was approved by the Ethics committee of the Polytechnic Institute of Santarém (approval number 102019Desporto).

### 2.1. Participants

One hundred and thirty white (Caucasian) children (67 boys, age 7.76 ± 0.12 years and 63 girls, age 7.59 ± 0.12 years) as determined by ”parents” place of birth, were recruited via word-of-mouth and through local school districts, all attending elementary schools in central and south of Portugal. To be included in the study, participants only needed to be apparently healthy without any contraindication to participate in a maximal CRF test. The exclusion criteria were a motor deficiency, medical contraindication to physical exercise or diagnosed disease or illness, or orthopedic issues that would limit their ability to run. During the collection period, children, parents, and teachers were asked not to perform any physical sports activities on the day before the tests. Although the initial sample was 140 children (70 boys and 70 girls), there was a 7% (*n* = 10) dropout because some children did not perform one of the tests (*n* = 4) or because the tests did not meet the eligibility criteria (*n* = 6).

### 2.2. Procedures

The study protocol involved three non-consecutive days for assessments. On the first day, the participants were certified regarding their health and clinical history by a technician in the presence of the classroom teacher. Body fat was determined with triceps and subscapular skinfold thickness measurements using a skinfold caliper, in a room with stable temperature and humidity (from 20 to 22 °C and from 50 to 60%, respectively). The second and third days were completed within a week, when participants completed two assessments of the 20mSRT at the same hour to avoid diurnal variation. All anthropometric and 20mSRT assessments were performed by the same certified fitness professional.

### 2.3. Anthropometric Assessment

During the first day, participants’ heights were measured and recorded to the nearest 0.1 cm, using a tape measure and the Frankfort plane procedures were applied [37,38]. Weight was measured to the nearest 100 g, using a standard scale (Beam Balance-Stadiometer model 700, SECA, Vogel & Halke, Hamburg, Germany). For body fat determination, triceps and subscapular skinfolds were measured using ISAK protocols with a skinfold caliper Slim Guide (Creative Health Products, Plymouth, MI, USA). The triceps skinfold (TricSKF) was measured on the back of the right arm over the triceps muscle, midway between the elbow and the acromion process of the scapula [39]. The subscapular skinfold (SubSKF) was measured 2 cm below the lower angle of the right scapula. A single evaluator assessed the anthropometric measurements, which were performed three times. Then, the average of the measurements was calculated [40,41]. All skinfolds were measured to the nearest 0.5 mm. The fat mass percentage was estimated using the Slaughter equation [41]. Specifically, the sum of the triceps and subscapular skinfold thicknesses < 35 mm was used for both girls and boys. The following formulas were applied:BF (%) = 1.21 (triceps and subscapular) − 0.008 (triceps and subscapular)2 − 1.7 (boys);
BF (%) = 1.33 (triceps and subscapular) − 0.013 (triceps and subscapular)2 − 2.5 (girls).

Then, the sum of the triceps and subscapular skinfold thicknesses > 35 mm was used for both girls and boys:BF (%) = 0.546 (triceps and subscapular) + 9.7 (boys);
BF (%) = 0.783 (triceps and subscapular) + 1.6 (girls).

As described by Lohman et al. [40], the waist circumference (WC) measurement was taken at the level of the umbilicus with a non-elastic flexible Rosscraft measuring tape (Rosscraft, Surrey, Canada) with a margin of error of 0.1 cm. A measure was taken with the subject standing without clothing covering the waist area.

### 2.4. mSRT Test Protocol

The 20mSRT is a progressive intensity test. The initial speed is 8.0 km/h, which is increased by 0.5 km/h in each stage after the first minute. For better clarity, each lap is followed by one beep, while each minute is followed by three beeps [42]. The protocol was conducted during the morning in sport pavilions at four schools in central and southern Portugal cities, during the spring (ambient temperature and percentage of relative humidity, respectively, from 20 to 24 °C and from 50 to 65%).

The standardized procedures of The Cooper Institute [43] for the 20mSRT were followed. The test was clearly explained to each participant, and all participants performed the test in their physical education classes. Before the application of protocol testing, a five-minute warm-up was performed with lower and upper limb stretching exercises and walking across the 20 m field test area.

As Scalco et al. [34] proposed during the test, all participants were verbally encouraged to achieve the best possible result. The types of verbal encouragement used were: “very well”, ”let’s go”, “way to go”, “you can do it”, “cheer up”, and “you are almost there”, spoken every 60 s, throughout the process, uninterrupted, and to all participants.

The test finished when a participant did not achieve, for the second time, the 20 m distance between two line within the specific time of the stage (controlled by the beep sounds). Finally, the total number of laps (TL) completed, and the speed reached at the end of the 20mSRT for each participant was used for analysis [43].

As mentioned, all participants performed two trials of the 20mSRT (within a week). During one trial, participants wore a face mask and a PGA for gas analysis, while in the other trial, they did not use them. A random order of the test with and without a face mask and PGA was applied.

### 2.5. Prediction of Peak Oxygen Consumption

Peak oxygen consumption (VO_2_peak) was measured by direct oximetry with a Cosmed K4*b*^2^ PGA analyzer (Cosmed, Rome, Italy), which had been previously validated [44]. In addition, the VO_2_peak values were also predicted by using 22 predictive equations (Table 1).

The cardiorespiratory data were collected as previously described by Silva et al. [47]. Specifically, the heart rate (HR) was measured using a Polar T31 sensor (Polar Electro Oy, Kempele, Finland) coupled to a Cosmed K4*b*^2^ analyzer. During each test, the HR and VO_2_ values of the participants were continuously monitored by telemetry. The size of the mask was adjusted to the child’s face, and the device’s harness was adjusted to the child’s trunk, carrying the portable unit in the chest area and the battery at the level of the shoulder blades. This compact device was attached without constricting the children’s movements. The K4*b*^2^ weighed 475 g, not exceeding ~1 kg of the total ”equipment” weight (harness and battery), and was not expected to significantly affect the energy demands of the subjects [52]. To assure the best fit and minimal dead space for pediatric testing, the mask chosen should be smaller or pediatric specific. After placing the mask, the mouthpiece was covered, and children were asked to make a forced expiration in order to check the sealing [53].

Before each use of the analyzer, calibration tests were performed [53], as described by the manufacturer, COSMED Srl (Rome, Italy). The gas analyzer calibration procedures before the start of each test were as follows: 45-min warm-up period for the device, calibration with ambient air, calibration with reference gas (16.7% O_2_ and 5.7% CO_2_), gas transition time calibration, and turbine calibration (with 3000 mL syringe) (Quinton Instruments, Seattle, WA, USA). Respiratory parameters were recorded breath by breath, and the values of VO_2_ were recorded for an average of 10 s [54,55].

There is no well-accepted definition of a VO_2_ plateau in pediatric testing, possibly because it is often absent [56,57,58]. Therefore, performance on the 20mSRT was considered to be maximal when the participant achieved at least two of the following adopted physiological criteria and one subjective criterion as proposed in a previous study by [59]. The physiological criteria included: exceeding the 2nd ventilatory threshold, interrupted as the maximal exhaustion test; respiratory exchange ratio (RER) score ≥1.0 [60]; VO_2_peak with the highest VO_2_ in mL/kg/min elicited during a progressive exercise test to exhaustion; reaching an aged-predicted maximal heart rate of ±10 bpm (Tanaka equation, 208–0.7 × age) [61]. The subjective criteria included: signs of maximal effort such as profuse sweating, facial flushing, and unsteady gait during the run [62]. Participants who did not achieve the previous criteria were excluded from the analysis (*n* = 6).

### 2.6. Statistical Analysis

The measured VO_2_peak values, the predicted VO_2_peak values using the 22 equations (Table 1), and all interest variable means and standard deviations (SD) were calculated for the total sample and divided by gender. Mean differences and confidence intervals at 95% (CI, 95%) were calculated for the comparisons between measured VO_2_peak values and those predicted using equations. Kolmogorov–Smirnov and the Levene tests were used to test the assumption of normality and homoscedasticity, respectively. Gender influences on the anthropometric, metabolic, cardiovascular, and performance variables from the 20mSRT and equations were determined with an independent *t*-test. Means comparisons between performance variables using, or not using, the K4*b*^2^ PGA were performed with paired sample *t*-tests, in the total population and between genders. A paired sample *t*-test was performed to estimate mean differences between measured VO_2_peak values and predicted VO_2_peak values using predictive equations. Correlations were determined between measured VO_2_peak values and predicted VO_2_peak values using equations, with and without the K4*b*^2^ PGA. The Sigmaplot version 14.0 software (Systat Software, San Jose, CA) was used to create the Bland–Altman plots. The graphical dispersion arrangement of Bland and Altman [63] allowed the visualization of the mean differences and the upper and lower limits according to two standard deviations of the differences in the measurements. The significance level considered for all tests was *p* < 0.05. All predictive equations were adjusted to fat mass (%). The statistical analyses were computed and performed using IBM SPSS Statistics for Windows, Version 27.0. (IBM Corp., Armonk, NY, USA). Hedge’s g effect size was also calculated for comparisons between genders, while Cohen’s d was calculated for the comparisons between use of the PGA and non-use of the PGA. The Hopkins’ thresholds for effect size statistics were used, as follows: ≤0.2, trivial; >0.2, small; >0.6, moderate; >1.2, large, >2.0, very large; and >4.0, nearly perfect [64].

## 3. Results

Table 2 presents the means and SDs of anthropometric, performance, metabolic, and cardiovascular variables assessed during the 20mSRT. There were differences between boys and girls in the tricipital skinfold (t(128) = 2.187, *p* < 0.05 and g = 0.384) and in all performance variables: MS with PGA (t(103.364) = −3.406, *p* < 0.001 and g = 0.661); MS without PGA (t(115.743) = −3.686, *p* < 0.001 and g = 0.674); TL with PGA (t(104.408) = −3.538, *p* < 0.001 and g = 0.610); TL without PGA (t(112.288) = −3.976, *p* < 0.001 and g = 0.689); TT with PGA (t(108.866) = −3.687, *p* < 0.001 and g = 0.638); TT without PGA (t(115.869) = −4.203, *p* < 0.001 and g = 0.729); and VO_2_peak (t(125.632) = −2.749, *p* < 0.05 and g = 0.48).

Table 3 presents differences between performance variables of the 20mSRT (maximal speed, total laps, and total time with and without the PGA).

Table 4 presents the 22 VO_2_peak predictive equations reported in the literature. Since many of the equations use gender as a variable, they were compared with themeasured VO_2_peak values from our study for the total sample and both genders. 

Table 5 presents the mean differences and comparisons between the measured VO_2_peak values and the predicted VO_2_peak values (highlighted with the letter “a”) for each predictive equation for girls, boys, and overall participants (highlighted with the letter “b”).

Table 6 presents only one non-significant VO_2_peak predictive equation, since they are the valuable equations to be discussed in the total sample. The SEE values ranged between 2.24 (Equation #4) and 7.09 mL.kg^−1^.min^−1^ (Equation #21, for the total sample). The validation coefficients (correlation between estimated and measured VO_2_peak) were significant for all equations (0.927 > r > 0.618, *p* < 0.001; and one presented 0.286 > r > 0.283, *p* < 0.05). In Table 6 and Table 7, the values are adjusted for fat mass (%).

Table 6 presents the limits of agreement (LoA) and range (upper LoA–lower LoA) for the entire sample. Equation #4 presents the smallest SEE value between the measured and estimated VO_2_peak values in the two tests with and without a PGA. However, it also presents the highest slope, meaning that the equation overpredicts the VO_2_peak in participants with lower VO_2_peak and underpredicts the VO_2_peak in participants with higher VO_2_peak.

Equation #21 presents the lowest slope but has the lowest R and the lowest range. Equation #1 presents a reasonably large d (*p* < 0.05), the highest range, and the highest slope (*p* < 0.0001).

Table 7 presents only the non-significant VO_2_peak predictive equations since they are the valuable equations to be discussed for both genders. Bland–Altman graphs were plotted to examine the bias distribution and assess the agreement between the measured VO_2_peak values and the predicted VO_2_peak values.

The Bland–Altman plots provide the systematic bias and random error between the measured and predicted VO_2_peak values, which are presented in Figure 1. Positive linear distributions of Equation #1 (Figure 1A,B), Equation #4 (Figure 1C,D), and Equation #6 (Figure 1E) are found. Equation #21 presents a more random dispersion of scores (Figure 1F,G).

## 4. Discussion

In this study, we aimed: (i) to verify if carrying and using a PGA constrains the performance of elementary children who perform the 20mSRT, (ii) to verify which of the VO_2_peak predictive equations have greater validity as compared with VO_2_peak values measured by direct oximetry using the the 20mSRT, (iii) to analyze if fat mass influences the results of predictive equations for VO_2_peak. The main findings showed that the MS, TL, and TT were higher for the boys than the girls, with and without the PGA. In addition, higher values were reported with the PGA than without the PGA for both girls and boys (except for boys in TT). Moreover, among the 22 predictive equations, only six equations were considered to estimate VO2peak correctly.

To discuss all objectives of the study, this section is organized into three subsections: (1) Constraints associated with using a portable gas analyzer; (2) agreement of VO2peak predictive equations as compared with VO_2_peak measured by direct oximetry; and (3) fat mass influence on the results of predictive equations for VO_2_peak.

### 4.1. Constraints Associated with Using a Portable Gas Analyzer

In this study, differences were found in the 20mSRT performance variables (maximal speed, total laps, and total time) when using a PGA versus without a PGA, for the entire sample. Interestingly, only the group of girls presented differences when performing for both genders. Specifically, without the PGA, MS was lower, TL were fewer, and TT was less for girls than for boys with the PGA. Based on the HRpeak, the participants of the present study showed values that were similar to those of other studies [30,35]. Task commitment and, eventually, motivational aspects can explain these results; however, the technicians gave proper verbal feedback and encouragement during testing [43].

To the best of our knowledge, only two studies have analyzed the constraints of using a PGA for assessing CRF in an infant and youth population. Selvadurai et al. [30] assessed 93 children and adolescents with cystic fibrosis, through the 20mSRT with and without a PGA (Cardiovit 100 CS Spirometry Module) but supported on a trolley placed on tracks and pushed by a technician. No differences in the cardiorespiratory responses were found (respiratory rate, Borg scale values, and peripheral oxygen saturation), as well as in the distance traveled, with or without using a PGA. The authors justified the results due to a light mask that weighed less than 100 g, which may have minimized any discomfort. However, their results were not in agreement with the results reported by Itaborahy et al. [35], who analyzed the reproducibility of intergroup performances performing a modified shuttle walk test, a 6-min walk test, and an ADL-Glittre for pediatric populations. They found reproducible results among themselves, but they were not reproducible when performed with or without a PGA (Cosmed K4*b*^2^, Cosmed, Rome, Italy). However, the authors stated that the performances in the 6-min walk and modified shuttle walk tests were not significantly different with or without a PGA.

### 4.2. Agreement of VO_2_peak Based on Predictive Equations as Compared with VO_2_peak Measured by Direct Oximetry

Accuracy is necessary to establish associations between cardiorespiratory fitness and other explored variables. The validity of previously published equations (22 equations) has been verified in Portuguese children. The main finding of our study was that, in 6–9-year-old children, different predictive equations presented variations for predicting VO_2_peak through the 20mSRT. Overall, predicted VO_2_peak values were only adequate in six equations (Equations #1, #3, #4, #6, #21 and Equation #12, only in boys). From the majority of the 22 existing predictive equations tested, the generalization and accuracy of the group or gender predicted VO_2_peak values seem to be unacceptable.

Regarding the total sample, a comparison of the predicted VO_2_peak values in the two 20mSRT trials (with and without a PGA) with the measured VO_2_peak values showed that, among the 22 equations, only Equations #1, #3, #4, #6 (but only in the trial performed without using the PGA), and #21 presented no differences. These results are in line with an evaluation carried out in a systematic review by [65], in which Equations #1, #6, and #7 had a strong level of evidence (equations validated by three or more studies with low risk of bias), and Equations #2, #3, and #4 were classified as moderate evidence. Menezes-Júnior et al. [65] also stated that Equation #6 presents higher evidence and reliability for predicting VO_2_peak values, for girls and boys.

In the present study, a comparison of the estimated VO_2_peak values in the two 20mSRT trials by gender showed that, in girls, only six of the 22 equations did not show differences (Equations #1, #4, #6, #7, #21, and #22). In boys, only seven of the 22 equations did not show differences (Equations #1, #3, #4, #6, #8, #12, and #21).

Although Equation #18 was developed using oxygen consumption data collected while the participants completed the 20mSRT, its predictive power did not prove to be accurate. Another study by [24] analyzed the VO_2_peak of Portuguese children while performing the 20mSRT with a PGA as compared with the predicted VO_2_peak values from the FITNESSGRAM reports and other equations [18,20,26,43]. They found that the FITNESSGRAM software was not significantly different from directly measured VO_2_peak values [24].

### 4.3. Fat Mass Influence on the Results of Predictive Equations for Predicting VO_2_peak

The different parameters that integrate the equations can give greater or lesser predictive power. Body mass index (BMI) has been considered to be the main body size predictor of VO_2_peak in several studies [18,27,51], indicating significantly improved performance of the published equations. The BMI can have a wide influence on CRF fitness in children and adolescents [16] and has a robust association with body mass. Some studies [18,24,43] have suggested that BMI or skinfold thickness, age, and gender can provide the best predicitons of VO_2_peak as compared with those that only include age. However, this study showed that the equations that only include age by Leger et al. [42] for the entire sample and by gender and the equation by Menezes-Junior et al. [1] for boys only presented reasonable predictions of VO_2_peak.

In addition, it has been reported that, in children between 11 and 17 years old, fat-free mass was the variable that affected VO_2_peak. Recently, some authors [27] have stated that the proportion-to-body mass scale exercise variables assume an underlying set of specific statistical assumptions that are rarely met in pediatric exercise research [2].

In this sense, a recent study by Menezes-Junior et al. [1] used the z-score of the BMI (BMI-z) in their equation and reported findings that suggested equations with the BMI-z and its combination with %FM were more accurate and suitable for predicting VO_2_peak values in boys. In the aforementioned study [1], the inclusion of %FM indicated improvements in the models, especially in boys. In contrast with our study, the Menezes-Junior [1] equation (that integrated the %FM) was only predictive for VO2peak in girls (6–9 years old). Despite the difference between the mean age of the sample in the present study (7.68 ± 0.08 years) as compared with the Menezes-Junior study [1] (13.37 ± 1.84 years), it is possible that the higher %FM in girls, which increases during puberty, may reduce their aerobic performance [21,66]. In contrast, the negative impact on boys’ CRF caused by an increase in body fat may have resulted from an unfit lifestyle that was not considered in the present analysis.

Despite using the BMI to predict VO_2_peak, Mahar et al. [50] considered the 20mSRT performance and age sufficient to predict VO_2_peak. However, they did not consider that measures of adiposity other than BMI had higher correlations with VO_2_peak. For instance, the correlation of FM with VO_2_peak was r = −0.38, while the correlation of BMI with VO_2_peak was r = −0.22 [50]. A study by Ayala-Guzmán and Ortiz-Hernández [21] showed the same tendency: VO_2_peak had a correlation of r = −0.61 with FM as compared with BMI (r = −0.22).

On the one hand, in the present study, only Equation #2 considered the gender and triceps skinfold, nevertheless, it was not predictive of VO_2_peak in the present sample. On the other hand, several equations have included gender influence in their equations [1,20,21,24,45,46,47,48,51]. Our results reported significant changes in performance tests regarding gender and, more precisely, in girls.

Some studies have observed that body fat displayed a better factor in the equations than BMI [21]. Nevertheless, in the present study, two equations, Equations #2 and #21 that integrated fat mass, were not found to be good predictors of VO_2_peak. Considering that excessive fat mass can decrease cardiorespiratory performance [21,67], this variable is suggested for better explanation of variations in VO_2_peak [1].

In opposition to the previous suggestion, in this study, we showed that Equation #21 presented a good prediciton of VO_2_peak using the interaction of height and age. One justification for this result could be the fact that older and taller adolescents may have an advantage performing the 20mSRT. However, such analysis was not considered in the present research.

Another study also assessed the agreement between VO_2_peak directly measured through the 20mSRT and VO_2_peak predicted using five different equations, in 13–19-year-old children [28], and from those equations, seven equations were used in our study, namely, Equations #1, #2, #3, #4, #6, #7, and #8. In a study by Ruiz et al. [28], the 20mSRT was performed by boys and girls with a mean age older than in the present study (girls, 14.6 ± 1.5 years and boys, 15.0 ± 1.6 years). All participants wore the K4*b*^2^ Cosmed PGA and measured VO_2_peak values of 47.1 ± 8.1 mL/kg/min, which was slightly lower than the values of our study, as well as the corresponding means of predicted VO_2_peak values of 41.5 ± 5.2 mL/kg/min for Equation #1, 44.2 ± 5.6 mL/kg/min for Equation #4, 45.7 ± 5.0 mL/kg/min for Equation #2, and 43 ± 5.5 mL/kg/min for Equation #7.

Most equations have been developed from data collected during a maximal treadmill protocol. Although previous research has shown that VO_2_peak values measured during the 20mSRT and treadmill were not statistically different [68], the treadmill involved continuous walking or running, while the 20mSRT test consisted of intermittent bouts of 20 m laps. The 20mSRT test is more similar to the sporadic, varying intensity activity patterns that youth typically engage in. As most equations were developed using data obtained from treadmill assessments, the accuracy of the VO_2_peak estimations should be similar across equations. However, the results from this study showed that the most accurate of all 22 equations were: Equations #1, #3, #4, #6 (but only in the trial performed without using the PGA), and #21.

According to the literature, Equation #1 was the first to predict VO_2_peak values in children and adolescents [42] and it has presented substantial evidence in a recent study [1]. In this study, Equations #1, #4, and #6 presented a strong association between the predicted VO_2_peak values and the measured VO_2_peak values. As compared with Menezes-Junior et al. [1], this study presented lower SEE values in the aforementioned equations. In addition, as compared with the study by Ruiz et al. [19], Equation #1 had a lower value in our study (respectively, 4.27 vs. 2.52 mL/kg/min). Despite this, Equation #21 had a lower correlational value, and high SEE value (with and without PGA, respectively, 7.07 and 7.09 mL/kg/min), which may be a potential equation to predict VO_2_peak values in girls, boys, and overall sample, since no significant differences were found between the measured and predicted VO_2_peak values for both conditions.

The Bland–Altman plots presented in this study (Figure 1) showed evidence of differences between the measured VO_2_peak and the predicted VO_2_peak values. It did not appear to consistently under/overestimate in the unfit/fit children with Equation #21, whereas this trend was evident with Equations #1, #4, and #6. For those, it can be said that for lower VO_2_peak values, the equation tends to overestimate the measured values, while for higher VO_2_peak values, the equation tends to underestimate the measured values. Only the plots regarding Menezes-Junior et al. [1] showed a random error which suggested a normal distribution of the differences between the methods, while the other plots presented a positive linear distribution of the values.

Possible explanations for the better validity of some equations could be related to the performance variables from 20mSRT, such as the number of laps, while others used the final speed or stage data, which were recorded only every minute of the test [18,19,20,42]. Thus, when the 20mSRT was stopped moments after the close of a stage or speed advance, only previous data were considered [69]. However, another study showed equivalent results for equations that included lap number or maximal velocity. Nevertheless, the authors recommended using lap numbers as a practical application [21].

The lack of accuracy between our collected data and the predicted values using the different predictive equations may be limited by the race/ethnic makeup of the sample groups, the lack of overweight and obese subjects, and the average age. Another limitation may be the convenience sampling techniques used, which may have elicited selection bias. Finally, sleep patterns were not controlled on the previous days of the CRF tests, which may have possibly interfered with the results. These factors can limit the generalizability of the results provided by the different equations.

Despite the previous limitations, the present study has several strengths. For instance, no allergic reactions were reported concerning the use of a PGA. In addition, there was a tendency for better performances for both the girls and the boys when using a PGA. Finally, the present study provides all equations that are better for predicting VO_2_peak values.

## 5. Conclusions

In this study, we found differences in the 20mSRT performance variables when using a PGA versus without a PGA, for the entire sample and for girls, where higher values of maximal speed, total laps, and time were found with the use of the PGA/K4*b*^2^. Thus, the first hypothesis was rejected since the PGA did not constrain such variables for girls while the opposite was observed for boys.

The reported data further indicate that, in 6–9-year-old children, predicted VO_2_peak values from the 20mSRT can vary using different predictive equations. For instance, predictions of VO_2_peak values were only adequate in six equations: #1 [42], #3 and #4 [20], #6 [18], #21 [1], and #12 only in boys [47]. Therefore, the second hypothesis of the present study was confirmed, and thus, suggested that these equations could be practically used to examine CRF in schools and in research with large populations when direct measurement of VO_2_peak is not feasible.

Finally, the third hypothesis was rejected, since fat mass did not show a stronger influence on the abovementioned six most adequate equations for predicitng VO_2_peak values from the 20mSRT.

For future research, it is suggested that this study be replicated with a larger sample for each age and with other brands of PGA.

## Figures and Tables

**Figure 1 biology-11-01356-f001:**
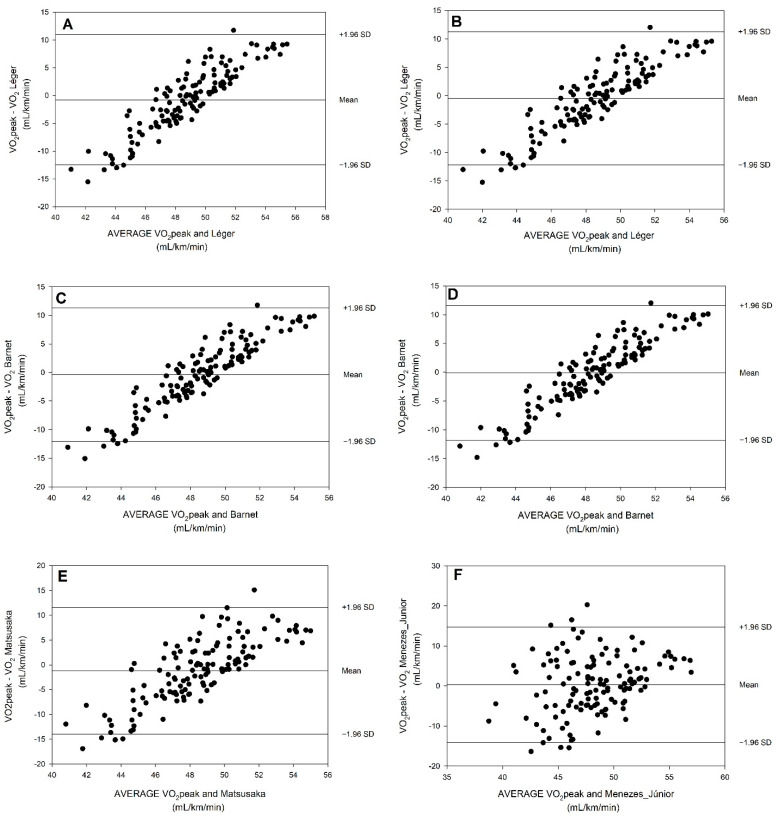
Bland–Altman plots regarding the measured VO2peak values and the predictive equations predicted VO2peak values: (**A**) Leger et al. (1988), Equation #1, with PGA; (**B**) Leger et al. [42], Equation #1, without PGA; (**C**) Barnett et al. [20], Equation #4, with PGA; (**D**) Barnett et al. [20], Equation #4, without PGA; (**E**) Matsuzaka et al. [18], Equation #6, with PGA; (**F**) Menezes-Junior et al. [1], Equation #21, with PGA; (**G**) Menezes-Junior et al. [1], Equation #21, without PGA.

**Table 1 biology-11-01356-t001:** VO_2_ peak predictive equations used with data from children and youth performing the 20mSRT.

NR	Equations	References
^#1^	VO_2_peak = 31.025 + (3.238 × SPEED) − (3.248 × AGE) + (0.1536 × AGE × SPEED), SPEED, maximal speed (km/h); AGE, years	Leger et al. [42]
^#2^	VO_2_peak = 28.3 − (2.1 × GENDER) − (0.7 × TRI_SK) + (2.6 × MS), GENDER, M = 0/F = 1; TRI_SK, tricipital skinfold (cm); MS, maximal speed (km/h)	Barnett et al. [20]
^#3^	VO_2_peak = 25.8 − (6.6 × GENDER) − (0.2 × BM) + (3.2 × MS), GENDER, M = 0/F = 1; BM, body mass (kg); MS, maximal speed (km/h)	
^#4^	VO_2_peak = 24.2 − (5.0 × GENDER) − (0.8 × AGE) + (3.4 × MS), GENDER, M = 0/F = 1; AGE, years; MS, maximal speed (km/h)	
^#5^	VO_2_peak = (0.35 × TL) − (0.59 × BMI) − (4.61 × GENDER) + 50.6, TL, total laps (no); BMI, body mass index (kg/m^2^); GENDER, M = 1/F = 2	Fernhall et al. [45]
^#6^	VO_2_peak = 25.9 − (2.21 × GENDER) − (0.0449 × AGE) − (0.831 × BMI) + (4.12 × MS), GENDER, M = 1/F = 0; AGE, years; BMI, body mass index (kg/m^2^); MS, maximal speed (km/h)	Matsuzaka et al. [18]
^#7^	VO_2_peak = 61.1 − (2.20 × GENDER) − (0.462 × AGE) − (0.862 × BMI) + (0.192 × TL), GENDER, M = 0/F = 1; BMI, body mass index (kg/m^2^); TL, Total laps (no)	
^#8^	VO_2_peak = 47.438 + (PACER × 0.142) + (GENDER × 5.134) − (BM × 0.197), PACER, maximal speed (km/h); GENDER, M = 1/F = 0; BM, body mass (kg)	Mahar et al. [46]
^#9^	VO_2_peak = 50.945 + (PACER × 0.126) + (GENDER × 4.946) − (BM × 0.655), PACER, maximal speed (km/h); GENDER, M = 1/F = 0; BM, body mass (kg)	
^#10^	^10^ VO_2_peak = (1/(1 + exp( − (1/(1 + exp(−((A1 ×0.8 + (−0.7)) × (−1.03329) + (B1 × 0.114285714286 + (−1.38571428571)) × 0.54719 + (C1 × 0.012213740458 + (−0.406870229008)) × 0.61542 + (D1 × 0.0195598978221 + (−2.76356892177)) × −0.51381 + (E1 × 0.0842105263158 + (−0.0684210526316)) × (−0.92239) + (−0.34242)))) × (−0.95905) + 1/(1 + exp(−((A1 × 0.8 + (−0.7)) × (−1.19367) + (B1 × 0.114285714286 + (−1.38571428571)) × (−1.54924) + (C1 × 0.012213740458 + (−0.406870229008)) × (−3.18931) + (D1 × 0.0195598978221 + (−2.76356892177)) × 0.77773 + (E1 × 0.0842105263158 + (−0.0684210526316)) × 3.31887 + (−0.55696)))) × 2.19501 + 1/(1 + exp(−((A1 × 0.8 + (−0.7)) × 1.38191 + (B1 × 0.114285714286 + (−1.38571428571)) × (−2.14449) + (C1 × 0.012213740458 + (−0.406870229008)) × 0.0485 + (D1 × 0.0195598978221 + (−2.76356892177)) × 0.10879 + (E1 × 0.0842105263158 + (−0.0684210526316)) × (−4.90052) + 0.53905))) × (−2.567) + (−0.05105)))) − (−0.478945173945))/0.0204587840012, A1, boys = 1/girls = 2; B1, age (years); C1, weight (kg); D1, height (cm); E1, total time (min)	Ruiz et al. [19]
^#11^	VO_2_peak = 41.76799 + (0.49261 × PACER) − (0.00290 × PACER ^ 2) − (0.61613 × BMI) + (0.34787 × GENDER × AGE), PACER, maximal speed (km/h); BMI, body mass index (kg/m^2^); GENDER, boy = 1/girl = 0	Mahar et al. [26]
^#12^	^12^ VO_2_peak = 43.313 + 4.567 × GENDER − 0.560 × BMI + 2.785 × STAGE, GENDER, boys = 1/girls = 0; BMI, body mass index (kg/m^2^); STAGE, Total time (min)	Silva et al. [47]
^#13^	VO_2_peak = (1/(1 + EXP(−((1/(1 + EXP(−(+((STAGE)/11) × −5.309 + (GENDER) × −1.968 + ((AGE − 10)/8) × 4.394 + ((HEIGHT −141)/46) × 1.881 + ((WEIGHT − 37)/59) × 3.078 + ((BMI − 16.23)/13.68) × 4.429 − 4.302)))) × −1.782 + (1/(1 + EXP(−(+((STAGE)/11) × 1.790 + (GENDER) × 2.253 + ((GENDER − 10)/8) × 1.770 + ((HEIGHT − 141)/46) × -1.060 + ((WEIGHT − 37)/59) × 4.978 + ((BMI − 16.23)/13.68) × − 3.610 − 2.705)))) × 9.988 + (1/(1 + EXP(−(+((STAGE)/11) × 5.528 + (GENDER) × −6.357 + ((AGE − 10)/8) × − 1.068 + ((HEIGHT − 141)/46) × 0.663 + ((WEIGHT − 37)/59) × 1.333 + ((BMI − 16.23)/13.68) × 0.825 − 1.608)))) × 6.384 + (1/(1 + EXP(−(+((STAGE)/11) × 8.144 + (GENDER) × −0.724 + ((AGE − 10)/8) × −0.329 + ((HEIGHT − 141)/46) × 6.170 + ((WEIGHT − 37)/59) × − 0.573 + ((BMI − 16.23)/13.68) × 0.373 − 4.679)))) × −4.278 − 3.886)))) × 39.83 + 29.17, STAGE, total laps; GENDER, boys = 1/girls = 0; AGE, years; HEIGHT, cm; WEIGHT, kg; BMI, body mass index (kg/m^2^)	
^#14^	VO_2_peak = 19.66 + (2.21 × MS) + (0.05 × AGE) + (2.08 × GENDER) − (0.38 × BMI), MS, maximal speed (km/h); AGE, years; GENDER, F = 0/M = 1; BMI, body mass index (kg/m^2^)	Quinart et al. [48]
^#15^	VO_2_peak = 45.619 + (0.353 × PACER) − (1.121 × AGE), PACER, total laps (no); AGE, years	Burns et al. [49]
^#16^	VO_2_peak = 44.862 + (0.347 × PACER) − (1.050 × AGE), PACER, total laps (no); AGE, years	Mahar et al. [50]
^#17^	VO_2_peak = 49.367 + (PACER × 0.331) − (AGE × 0.777) − (BMI × 0.369), PACER, total laps (no); AGE, years; BMI, body mass index (kg/m^2^)	
^#18^	VO_2_peak = 52.934 + 0.256 × (LAPS) − 0.924 × (BMI) + 0.468 × (GENDER × AGE), LAPS, total laps (no); BMI, body mass index (kg/m^2^); Gender, F = 0/M = 1; AGE, years	Scott et al. [51]
^#19^	VO_2_peak = 44.942 + (0.646 × AGE) − (6.586 × GENDER) + (0.318 × PACER) − (0.243 × WAIST_CIRC), AGE, years; GENDER, F = 1/M = 0; PACER, total laps (no); WAIST_CIRC, waist circumference (cm)	Ayala-Guzman and Ortiz-Hernandez [21]
^#20^	VO_2_peak = 37.009 + (0.408 × AGE) − (4.759 × GENDER) + (0.284 × PACER) − (0.312 × BF), AGE, years; GENDER, F = 1/M = 0; PACER, total laps (no); BF, body fat (%)	
^#21^	VO_2_peak = 46.802 + (0.381 × LAPS) + (−3.682 × BMI-Z) + (−0.0568 × HEIGHT × AGE) + (3.078 × GENDER), LAPS, total laps (no); BMI-Z, body mass index (Z_score); HEIGHT, m; AGE, months; GENDER, F = 0/M = 1	Menezes-Junior et al. [1]
^#22^	VO_2_peak = 54.051 + (0.324 × LAPS) + (−2.626 × BMI-Z) + (−0.276 × %FM) + (−0.0493 × HEIGHT × AGE) + (2.016 × GENDER), LAPS, total laps (no); BMI-Z, body mass index (Z_score); %FM, fat mass (%); HEIGHT, m; AGE, months; GENDER, F = 0/M = 1	

NR, number of the equation.

**Table 2 biology-11-01356-t002:** Sample physical characteristics and metabolic, cardiovascular, and performance variables assessed during the 20mSRT (mean ± SD).

	Girls (*n* = 63)	Boys (*n* = 67)	Total (*n* = 130)
**Age (years)**	7.59 ± 0.12	7.76 ± 0.12	7.68 ± 0.94
**Anthropometric variables**			
Height (cm)	128.70 ± 7.52	130.14 ± 9.23	129.44 ± 8.45
Body mass (kg)	29.22 ± 6.14	31.02 ± 8.87	30.15 ± 7.70
Fat mass (%)	16.92 ± 5.10	15.59 ± 6.64	16.23 + 5.96
Body mass index (kg/m^2^)	17.50 ± 2.51	17.97 ± 2.78	17.74 ± 2.65
Body mass index z-score (kg/m^2^)	0.53 ± 0.93	0.73 ± 0.88	0.63 ± 0.91
Waist circumference (cm)	56.72 ± 6.19	58.95 ± 8.98	57.87 ± 7.80
Tricipital skinfold (cm)	11.70 ± 3.96 *	10.11 ± 4.33	10.88 ± 4.22
Subscapular skinfold (cm)	6.76± 2.82	6.39 ± 3.33	6.57 ± 3.13
**Metabolic and cardiovascular variables**			
Heart rate peak (bpm)	193.42 ± 9.0	194.09 ± 8.19	193.77 ± 8.56
VO_2_peak (ml/kg/min)	46.97 ± 5.07 *	49.69 ± 6.19	48.37 ± 5.81
**Performance variables from the multistage 20-m shuttle run test**			
Maximal speed with PGA (km/h)	9.55 ± 0.42 **	9.92 ± 0.77	9.74 ± 0.65
Maximal speed without PGA (km/h)	9.44 ± 0.54 **	9.88 ± 0.81	9.67 ± 0.73
Total laps with PGA (nr)	19.92 ± 7.03 **	26.22 ± 12.66	23.17 ± 10.76
Total laps without PGA (nr)	18.46 ± 8.19 **	25.99 ± 12.99	22.34 ± 11.52
Total time with PGA (s)	159.86 ± 52.21 **	206.12 ± 87.45	183.70 ± 75.90
Total time without PGA (s)	149.96 ± 62.00 **	207.76 ± 92.64	179.75 ± 84.14

PGA, portable gas analyzer; nPGA, without PGA; nr, number; s, seconds; * difference between girls and boys (*p* < 0.05); ** difference between girls and boys (*p* < 0.001).

**Table 3 biology-11-01356-t003:** Comparison of the 20mSRT performance variables in the two trials with and without the portable gas analyzer.

	Girls (*n* = 63)		Boys (*n* = 67)		Total (*n* = 130)	
	Diff (CI 95%)	PGA vs. nPGA	Diff (CI 95%)	PGA vs. nPGA	Diff (CI 95%)	PGA vs. nPGA
MS-PGA (km/h) − MS-nPGA (km/h)	0.11 (0.03–0.20)	*p* = 0.010; d = 0.23	0.04 (−0.03–0.11)	*p* = 0.301; d = 0.05	0.07 (0.02–0.13)	*p* = 0.009; d = 0.10
TL-PGA (nr) − TL-nPGA (nr)	1.46 (0.54–2.38)	*p* = 0.002; d = 0.19	0.24 (−0.61–1.09)	*p* = 0.557; d = 0.02	0.83 (0.20–1.46)	*p* = 0.010; d = 0.07
TT-PGA (s) − TT-nPGA (s)	9.90 (2.10–17.69)	*p* = 0.014; d = 0.17	−1.64 (−8.47–5.20)	*p* = 0.634; d = −0.02	3.95 (−1.24–9.14)	*p* = 0.135; d = 0.05

PGA, portable gas analyzer; nPGA, without PGA; MS-PGA, maximal speed with PGA; MS-nPGA, maximal speed without PGA; TL-PGA, total laps with PGA; TL-nPGA, total laps without PGA; TT-PGA, total time with PGA; TT-nPGA, total time without PGA; nr, number; s, seconds; Diff, mean difference; CI, confidence interval.

**Table 4 biology-11-01356-t004:** VO_2_peak measured and VO_2_peak predicted by equations regarding the total sample and genders during the 20mSRT (mean ± SD in mL/kg/min).

	Girls (*n* = 63)		Boys (*n* = 67)			Total (*n* = 130)	
	Mean ± SD	PGA vs. nPGA	Mean ± SD	PGA vs. nPGA	Girls vs. Boys	Mean ± SD	PGA vs. nPGA
**VO_2_peak measured**	46.97 ± 5.07		49.69 ± 6.19		*p* = 0.007; g = 0.48	48.37 ± 5.81	
Leger et al. [42]	^1^ 48.46 ± 1.09 ^a^	* p * = 0.012 d = 0.27	^1^ 49.81 ± 2.98 ^a*^	* p * = 0.306 d = 0.05	* p * = 0.002; g = 0.59	^ 1 ^ 49.15 ± 2.60 ^a^	* p * = 0.010 d = 0.12
	^1^ 47.97 ± 2.25 ^b^		^1^ 49.65 ± 3.17 ^b*^		* p * = 0.001; g = 0.61	^ 1 ^ 48.83 ± 2.88 ^b^	
Barnett et al. [20]	^2^ 42.84 ± 3.30 ^a^	* p * = 0.010 d = 0.09	^2^ 47.01 ± 4.11 ^a^	* p * = 0.301 d = 0.02	* p * < 0.001; g = 1.12	^ 2 ^ 44.99 ± 4.27 ^a^	* p * = 0.009 d = 0.05
	^2^ 42.54 ± 3.41 ^b^		^2^ 46.91 ± 4.11 ^b^		* p * < 0.001; g = 1.15	^ 2 ^ 44.79 ± 4.37 ^b^	
	^3^ 43.92 ± 1.96 ^a^	* p * = 0.010 d = 0.18	^3^ 51.33 ± 3.23 ^a^	* p * = 0.301 d = 2.75	* p * < 0.001; g = 2.75	^ 3 ^ 47.74 ± 4.59 ^a^	* p * = 0.009 d = 0.05
	^3^ 43.55 ± 2.12 ^b^		^3^ 51.74 ± 3.29 ^b^		* p * < 0.01; g = 2.94	^ 3 ^ 47.50 ± 4.74 ^b^	
	^4^ 45.60 ± 1.31 ^a^	* p * = 0.010 d = 0.26	^4^ 51.71 ± 2.33 ^a^	* p * = 0.301 d = 0.05	* p * < 0.001; g = 3.21	^ 4 ^ 48.75 ± 3.61 ^a^	* p * = 0.009 d = 0.07
	^4^ 45.21 ± 1.66 ^b^		^4^ 51.59 ± 2.47 ^b^		* p * < 0.001; g = 3.01	^ 4 ^ 48.50 ± 3.83 ^b^	
Fernhall et al. [45]	^5^ 39.89 ± 4.08 ^a^	* p * = 0.004 d = 0.08	^5^ 42.82 ± 6.07 ^a^	* p * = 0.201 d = 0.04	* p * = 0.002; g = 0.56	^ 5 ^ 41.40 ± 5.39 ^a^	* p * = 0.010 d = 0.05
	^5^ 39.54 ± 4.30 ^b^		^5^ 42.58 ± 6.41 ^b^		* p * = 0.002; g = 0.55	^ 5 ^ 41.11 ± 6.68 ^b^	
Matsuzaka et al. [18]	^6^ 48.15 ± 3.14 ^a^	* p * = 0.010 d = 0.14	^6^ 51.48 ± 4.38 ^a^	* p * = 0.301 d = 0.03	* p * < 0.001; g = 0.87	^ 6 ^ 49.87 ± 4.16 ^a^	* p * = 0.09 d = 0.07
	^6^ 47.68 ± 3.39 ^b^		^6^ 51.33 ± 4.48 ^b^		* p * < 0.001; g = 0.91	^ 6 ^ 49.56 ± 4.37 ^b^	
	^7^ 45.48 ± 3.04 ^a^	* p * = 0.004 d = 0.06	^7^ 45.79 ± 4.14 ^a^	* p * = 0.201 d = 0.03	* p * = 0.632; g = 0.08	^ 7 ^ 45.64 ± 3.64 ^a^	* p * = 0.010 d = 0.04
	^7^ 45.29 ± 3.14 ^b^		^7^ 45.66 ± 4.30 ^b^		* p * = 0.583; g = 0.10	^ 7 ^ 45.48 ± 3.77 ^b^	
Mahar et al. [46]	^8^ 43.04 ± 1.22 ^a^	* p * = 0.010 d = 0.02	^8^ 47.87 ± 1.77 ^a^	* p * = 0.301 d = 0.01	* p * < 0.001; g = 3.16	^ 8 ^ 45.53 ± 2.86 ^a^	* p * = 0.09 d = 0.00
	^8^ 43.02 ± 1.21 ^b^		^8^ 47.86 ± 1.76 ^b^		* p * < 0.001; g = 3.19	^ 8 ^ 45.52 ± 2.86 ^b^	
	^9^ 33.01 ± 4.03 ^a^	* p * = 0.010 d = 0.00	^9^ 36.82 ± 5.83 ^a^	* p * = 0.301 d = 0.00	* p * < 0.001; g = 0.76	^ 9 ^ 35.00 ± 5.37 ^a^	* p * = 0.09 d = 0.01
	^9^ 33.00 ± 4.02 ^b^		^9^ 36.82 ± 5.82 ^b^		* p * < 0.001; g = 0.76	^ 9 ^ 34.97 ± 5.37 ^b^	
Ruiz et al. [19]	^10^ 32.07 ± 1.95 ^a^	* p * = 0.126 d = 0.09	^10^ 42.07 ± 6.68 ^a^	* p * = 0.391 d = 0.03	* p * < 0.001; g = 2.01	^ 10 ^ 37.22 ± 7.06 ^a^	* p * = 0.950 d = 0.00
	^10^ 31.87 ± 2.27 ^b^		^10^ 42.28 ± 6.96 ^b^		* p * < 0.001; g = 1.99	^ 10 ^ 37.23 ± 7.38 ^b^	
Mahar et al. [26]	^11^ 35.42 ± 1.62 ^a^	* p * = 0.010 d = 0.03	^11^ 35.29 ± 1.83 ^a^	* p * = 0.295 d = 0.01	* p * = 0.671; g = 0.08	^ 11 ^ 35.36 ± 1.73 ^a^	* p * = 0.08 d = 0.02
	^11^ 35.37 ± 1.62 ^b^		^11^ 35.28 ± 1.83 ^b^		* p * = 0.752; g = 0.05	^ 11 ^ 35.32 ± 1.72 ^b^	
Silva et al. [47]	^12^ 42.68 ± 3.89 ^a^	* p * = 0.124 d = 0.05	^12^ 45.74 ± 5.62 ^a^	* p * = 0.461 d = 0.03	* p * < 0.001; g = 0.63	^ 12 ^ 44.26 ± 5.08 ^a^	* p * = 0.135 d = 0.04
	^12^ 42.46 ± 4.32 ^b^		^12^ 45.59 ± 6.06 ^b^		* p * = 0.001; g = 0.59	^ 12 ^ 44.07 ± 5.50 ^b^	
	^13^ 34.69 ± 4.84 ^a^	* p * = 0.019 d = 0.06	^13^ 35.10 ± 6.04 ^a^	* p * =0.177 d = 0.06	* p * = 0.664; g = 0.07	^ 13 ^ 34.90 ± 5.47 ^a^	* p * = 0.023 d = 0.07
	^13^ 34.40 ± 4.61 ^b^		^13^ 34.73 ± 5.77 ^b^		* p * = 0.669; g = 0.06	^ 13 ^ 34.54 ± 5.22 ^b^	
Quinart et al. [48]	^14^ 34.49 ± 1.56 ^a^	* p * = 0.010 d = 0.15	^14^ 37.22 ± 2.24 ^a^	* p * = 0.301 d = 0.04	* p * < 0.001; g = 1.41	^ 14 ^ 35.90 ± 2.37 ^a^	* p * = 0.009 d = 0.07
	^14^ 34.24 ± 1.71 ^b^		^14^ 37.14 ± 2.24 ^b^		* p * < 0.001; g = 1.45	^ 14 ^ 35.73 ± 2.49 ^b^	
Burns et al. [49]	^15^ 44.15 ± 2.19 ^a^	* p * =0.02 d = 0.21	^15^ 46.18 ± 4.05 ^a^	* p * = 0.577 d = 0.02	* p * = 0.001; g = 0.62	^ 15 ^ 45.19 ± 3.43 ^a^	* p * = 0.010 d = 0.08
	^15^ 43.63 ± 2.67 ^b^		^15^ 46.09 ± 4.17 ^b^		* p * < 0.001; g = 0.70	^ 15 ^ 44.90 ± 3.72 ^b^	
Mahar et al. [50]	^16^ 43.81 ± 2.16 ^a^	* p * = 0.02 d = 0.21	^16^ 45.81 ± 3.40 ^a^	* p * = 0.577 d = 0.02	* p * = 0.001; g = 0.70	^ 16 ^ 44.84 ± 3.38 ^a^	* p * = 0.010 d = 0.08
	^16^ 43.30 ± 2.62 ^b^		^16^ 45.73 ± 4.11 ^b^		* p * < 0.001; g = 0.70	^ 16 ^ 44.55 ± 3.67 ^b^	
	^17^ 43.61 ± 2.56 ^a^	* p * = 0.02 d = 0.18	^17^ 45.39 ± 4.31 ^a^	* p * = 0.577 d = 0.00	* p * = 0.005; g = 0.50	^ 17 ^ 44.52 ± 3.67 ^a^	* p * = 0.010 d = 0.07
	^17^ 43.12 ± 2.96 ^b^		^17^ 45.31 ± 4.42 ^b^		* p * = 0.001; g = 0.58	^ 17 ^ 44.25 ± 3.92 ^b^	
Scott et al. [51]	^18^ 41.86 ± 3.30 ^a^	* p * = 0.02 d = 0.11	^18^ 46.67 ± 4.75 ^a^	* p * = 0.577 d = 0.01	* p * < 0.001; g = 1.17	^ 18 ^ 44.34 ± 4.75 ^a^	* p * = 0.010 d = 0.04
	^18^ 41.49 ± 3.53 ^b^		^18^ 46.61 ± 4.81 ^b^		* p * < 0.001; g = 1.21	^ 18 ^ 44.13 ± 4.95 ^b^	
Ayala-Guzman & Ortiz-Hernandez [21]	^19^ 35.81 ± 3.07 ^a^	* p * = 0.02 d = 0.14	^19^ 43.97 ± 5.30 ^a^	* p * = 0.577 d = 0.15	* p * < 0.001; g = 1.87	^ 19 ^ 40.02 ± 5.97 ^a^	* p * = 0.010 d = 0.04
	^19^ 35.35 ± 3.37 ^b^		^19^ 43.89 ± 5.40 ^b^		* p * < 0.001; g = 1.88	^ 19 ^ 39.75 ± 6.23 ^b^	
	^20^ 35.73 ± 2.93 ^a^	* p * = 0.02 d = 0.14	^20^ 42.76 ± 4.80 ^a^	* p * = 0.577 d = 0.01	* p * < 0.001; g = 1.76	^ 20 ^ 39.35 ± 5.33 ^a^	* p * = 0.010 d = 0.04
	^20^ 35.31 ± 3.24 ^b^		^20^ 42.69 ± 4.81 ^b^		* p * < 0.001; g = 1.79	^ 20 ^ 39.11 ± 5.53 ^b^	
Menezes-Junior et al. [1]	^21^ 45.76 ± 4.80 ^a^	* p * = 0.02 d = 0.11	^21^ 50.26 ± 6.41 ^a^	* p * = 0.577 d = 0.01	* p * < 0.001; g = 0.79	^ 21 ^ 48.08 ± 6.10 ^a^	* p * = 0.010 d = 0.05
	^21^ 45.21 ± 5.10 ^b^		^21^ 50.17 ± 6.50 ^b^		* p * < 0.001; g = 0.85	^ 21 ^ 47.77 ± 6.35 ^b^	
	^22^ 48.65 ± 4.81 ^a^	* p * = 0.02 d = 0.10	^22^ 52.34 ± 6.53 ^a^	* p * = 0.577 d = 0.01	* p * < 0.001; g = 0.64	^ 22 ^ 50.55 ± 6.03 ^a^	* p * = 0.010 d = 0.04
	^22^ 48.18 ± 5.07 ^b^		^22^ 52.26 ± 6.55 ^b^		* p * < 0.001; g = 0.69	^ 22 ^ 50.28 ± 6.20 ^b^	

^1, 2, 3, … 22^, Number of equations according to Table 1; PGA, portable gas analyzer; nPGA, without PGA; ^a^ calculation based on a test performed with PGA; ^b^, calculation based on test performed without PGA; d, Cohen’s d effect size; g, Hedge effect size.

**Table 5 biology-11-01356-t005:** Comparisons between the measured VO_2_peak values and predited VO_2_peak values using equations.

References	Equation Number *	Girls (*n* = 63)			Boys (*n* = 67)			Total (*n* = 130)		
		**Diff (CI 95%)**	** *p* **	**d**	**Diff (CI 95%)**	** *p* **	**d**	**Diff (CI 95%)**	** *p* **	**d**
Leger et al. [42]	1	−1.00 (−2.20–0.20)	0.101	0.25	0.041 (−0.99–1.07)	0.936	0.01	−0.46 (1.24–0.32)	0.243	0.10
Barnett et al. [20]	2	4.43 (3.44–5.42)	<0.001	1.03	2.78 (1.78–3.77)	<0.001	0.53	3.58 (2.87–4.28)	<0.001	0.70
	3	3.41 (2.38–4.45)	<0.001	0.88	−1.53 (−2.48–−0.57)	0.002	0.41	0.87 (0.05–1.68)	0.037	0.16
	4	1.75 (0.60–2.91)	0.004	0.47	−1.90 (−2.96–−0.83)	0.001	0.40	−0.13 (−0.96–0.71)	0.763	0.03
Fernhall et al. [45]	5	7.43 (5.81–9.04)	<0.001	1.58	7.11 (5.19–9.04)	<0.001	1.13	7.26 (6.02–8.51)	<0.001	1.16
Matsuzaka et al. [18]	6	−0.72 (−1.71–0.27)	0.153	0.16	−1.64 (−2.49–−0.78)	<0.001	0.30	−1.19 (−1.84–−0.54)	<0.001	0.23
	7	1.67 (0.21–3.14)	0.026	0.40	4.03 (2.34–5.73)	<0.001	0.76	2.89 (1.76–4.02)	<0.001	0.59
Mahar et al. [46]	8	3.95 (2.76–5.13)	<0.001	1.07	1.82 (.42–3.23)	0.012	0.40	2.85 (1.92–3.78)	<0.001	0.62
	9	13.97 (12.69–15.25)	<0.001	3.05	12.87 (11.19–14.55)	<0.001	2.14	13.40 (12.35–14.46)	<0.001	2.40
Ruiz et al. [19]	10	15.10 (14.03–16.17)	<0.001	3.84	7.41 (6.42–8.41)	<0.001	1.13	11.14 (10.16–12.12)	<0.001	1.68
Mahar et al. [26]	11	11.59 (10.50–12.69)	<0.001	3.08	14.41 (13.09–15.73)	<0.001	3.16	13.05 (12.16–13.93)	<0.001	3.05
Silva et al. [47]	12	4.51 (2.88–6.13)	<0.001	0.96	4.10 (2.21–5.99)	<0.001	0.67	4.30 (3.06–5.53)	<0.001	0.76
	13	12.63 (10.82–14.43)	<0.001	2.59	14.96 (12.82–17.10)	<0.001	2.50	13.83 (12.43–15.23)	<0.001	2.50
Quinart et al. [48]	14	12.72 (11.67–13.78)	<0.001	3.36	12.55 (11.47–13.63)	<0.001	2.70	12.64 (11.89–13.38)	<0.001	2.83
Burns et al. [49]	15	3.34 (2.21–4.46)	<0.001	0.82	3.60 (2.71–4.48)	<0.001	0.68	3.47 (2.77–4.17)	<0.001	0.71
Mahar et al. [50]	16	3.67 (2.55–4.79)	<0.001	0.91	3.96 (3.07–4.85)	<0.001	0.75	3.82 (3.12–4.52)	<0.001	0.79
	17	3.84 (2.81–4.88)	<0.001	0.93	4.38 (3.55–5.22)	<0.001	0.81	4.12 (3.47–4.77)	<0.001	0.83
Scott et al. [51]	18	5.48 (4.50–6.46)	<0.001	1.25	3.08 (2.21–3.94)	<0.001	0.56	4.24 (3.56–4.92)	<0.001	0.79
Ayala-Guzman and Ortiz-Hernandez [21]	19	11.62 (10.62–12.62)	<0.001	2.70	5.79 (4.98–6.61)	<0.001	1.00	8.62 (7.81–9.43)	<0.001	1.43
	20	11.66 (10.70–12.61)	<0.001	2.71	7.00 (6.20–7.80)	<0.001	1.26	9.26 (8.52–9.99)	<0.001	1.63
Menezes-Junior et al. [1]	21	1.76 (0.64–2.88)	0.003	0.25	−0.48 (−1.42–0.45)	0.305	0.08	0.60 (−0.14–1.35)	0.111	0.10
	22	−1.21 (−2.30–−0.12)	0.031	0.24	−2.57 (−3.59–−1.56)	<0.001	0.40	−1.91 (−2.66–−1.17)	<0.001	0.32

* The number of equations according to Table 1; Diff, mean difference in ml/kg/min; CI, confidence interval; d, Cohen’s d.

**Table 6 biology-11-01356-t006:** Predictive equations using a Bland–Altman approach as compared with measured VO_2_peak values (mL/kg/min).

Equations and References	LoA		Slope	R	R^2^	SEE
^#1, a, b^ Leger et al. [42]	−12.53	10.96 ^a^	1.837 **	0.907 **	0.822	2.53
	−12.20	11.28 ^b^	1.838 **	0.908 **	0.823	2.52
^#4, a, b^ Barnett et al. [20]	−12.06	11.30 ^a^	1.873 **	0.927 **	0.859	2.25
	−11.79	11.53 ^b^	1.875 **	0.927 **	0.860	2.24
^#6, b^ Matsuzaka et al. [18]	−14.01	11.62 ^b^	1.265 **	0.618 **	0.383	5.16
^#21, a, b^ Menezes-Junior et al. [1]	−14.12	14.69 ^a^	0.588 *	0.286 *	0.082	7.07
	−13.82	15.03 ^b^	0.582 *	0.283 *	0.080	7.09

*, **, difference between directly measured VO_2_peak and estimated VO_2_peak from each equation for, respectively, *p* < 0.05 and *p* < 0.001; ^a^, with PGA; ^b^, without PGA; LoA, limits of agreement and range (upper LoA–lower LoA) are reported in mL/kg/min. Predictive equations for VO_2_peak; ^#1^
VO_2_peak = 31.025 + (3.238 × SPEED) − (3.248 × AGE) + (0.1536 × AGE × SPEED), SPEED, maximal speed (km/h); AGE, years; ^#4^
VO_2_peak = 24.2 − (5.0 × GENDER) − (0.8 × AGE) + (3.4 × MS), GENDER, M = 0/F = 1; AGE, years; MS, maximal speed (km/h); ^#6^
VO_2_peak = 25.9 − (2.21 × GENDER) − (0.0449 × AGE) − (0.831 × BMI) + (4.12 × MS), GENDER, M = 1/F = 0; AGE, years; BMI, body mass index (kg/m^2^); MS, maximal speed (km/h); ^#21^
VO_2_peak = 46.802 + (0.381 × LAPS) + (−3.682 × BMI-Z) + (−0.0568 × HEIGHT × AGE) + (3.078 × GENDER), LAPS, total laps (no); BMI-Z, body mass index (Z_score); HEIGHT, m; AGE, months; GENDER, F = 0/M = 1.

**Table 7 biology-11-01356-t007:** Predictive equations using a Bland–Altman approach as compared with the measured VO_2_ values (ml/kg/min) in both genders.

	Girls					
Equations and references	LoA		Slope	R	R^2^	SEE
^#1, a, b^ Leger et al. [42]	−12.88	9.58 ^a^	2.043 **	0.924 **	0.851	2.21
	−12.55	9.90 ^b^	2.044 **	0.924 **	0.854	2.72
^#4, a, b^ Barnett et al. [20]	−12.39	9.86 ^a^	2.056 **	0.940 **	0.884	1.95
	−12.13	10.11 ^b^	2.056 **	0.914 **	0.885	1.94
^#6, a^ Matsuzaka et al. [18]	−14.44	10.68 ^b^	1.633 **	0.673 **	0.454	4.77
^#21, a, b^ Menezes-Junior et al. [1]	−14.42	13.97 ^a^	0.922 *	0.362 *	0.131	6.80
	−14.12	14.30 ^b^	0.915 *	0.359 *	0.003	4.61
	**Boys**					
^#1, a, b^ Leger et al. [42]	−12.04	12.10 ^a^	1.730 **	0.896 **	0.804	2.75
	−11.72	12.42 ^b^	1.731 **	0.897 **	0.805	2.74
^#4, a, b^ Barnett et al. [20]	−11.57	12.47 ^a^	1.778 **	0.918 **	0.843	2.45
	−11.32	12.73 ^b^	1.780 **	0.919 **	0.844	4.22
^#6, a^ Matsuzaka et al. [18]	−13.55	12.47 ^b^	1.088 **	0.581 **	0.338	3.17
^#21, a, b^ Menezes-Junior et al. [1]	−13.88	15.42 ^a^	0.427	0.231	0.039	7.33
	−13.58	15.75 ^b^	0.421	0.228 *	0.037	3.86

*, **, difference between directly measured VO_2_peak and estimated VO_2_peak from each equation for, respectively, *p* < 0.05 and *p* < 0.001; ^a^, with PGA; ^b^, without PGA; LoA, limits of agreement and range (upper LoA–lower LoA) are reported in mL/kg/min. Predictive equations for VO_2_peak; ^#1^
VO_2_peak = 31.025 + (3.238 × SPEED) − (3.248 × AGE) + (0.1536 × AGE × SPEED), SPEED, maximal speed (km/h); AGE, years; ^#4^
VO_2_peak = 24.2 − (5.0 × GENDER) − (0.8 × AGE) + (3.4 × MS), GENDER, M = 0/F = 1; AGE, years; MS, maximal speed (km/h); ^#6^
VO_2_peak = 25.9 − (2.21 × GENDER) − (0.0449 × AGE) − (0.831 × BMI) + (4.12 × MS), GENDER, M = 1/F = 0; AGE, years; BMI, body mass index (kg/m^2^); MS, maximal speed (km/h); ^#21^
VO_2_peak = 46.802 + (0.381 × LAPS) + (−3.682 × BMI-Z) + (−0.0568 × HEIGHT × AGE) + (3.078 × GENDER), LAPS, total laps (no); BMI-Z, body mass index (Z_score); HEIGHT, m; AGE, months; GENDER, F = 0/M = 1.

## Data Availability

The data presented in this study are available on request from the corresponding authors.

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
