# Peer review of "The Multistage 20-m Shuttle Run Test for Predicting VO2Peak in 6–9-Year-Old Children: A Comparison with VO2Peak Predictive Equations"

_biology, 2022, doi:10.3390/biology11091356_

Round 1
Reviewer 1 Report
The original research manuscript "The 20-m multistage-shuttle run-test to predict VO2peak in 6-9 year-old children. A comparison of VO2 peak equations" has important research significance. The manuscript is logically clear and well written. But it should be further refined before publication.
1. The significance of this study should be further explained in the preface.
2. The sampling process of the subjects should also be explained in detail. It is recommended to use a flowchart to explain the sampling process of the subjects more intuitively.
3. The discussion part can be appropriately reduced. For example, "4.1 Constraints of using the portable gas analyzer" does not seem to be the focus of the discussion, and unnecessary descriptions should be reduced.
Hopefully the above comments will lead to some better improvements to your manuscript.
Reviewer 2 Report
Thank you for the opportunity to review this manuscript. Overall, this is a well-written paper with an interesting result on sport area, with a direct application. The results are based on rational working objectives, well described and with a correct research design.
In my opinion this is an excellent work that need minimum modifications in order than I submit. The authors conducted a really good research.
INTRODUCTION
The introduction provides sufficient background information for readers to understand the research aim, however the authors should clarify the importance of the evaluation of VO2max in these ages.
Motivations for this study are more than clear and the objectives are clearly defined at the Introduction, the argumentation in this part was concise.
METHODS
The methodology proposed to reach the aim of the study looks appropriate, well designed and conducted. There are few instances where assertions are made that are not substantiated with references.
RESULTS
Results paragraph should include the most relevant data.
All of the tables and figures explain in a correct direction the data obtained
DISCUSSION
All possible interpretations of the data considered are consistent, however, there are some grammar mistakes that should be corrected.
The conclusion should respond to the research aim
Explain limitations of the study and future research line according to the study's conclusion
LITERATURE CITED
The literature cited is relevant to the study, but there are several instances in which the author makes assertions without substantiating them with references, but which are sustained by the main text and previous citations.
Reference style should be checked with the journal standards.
SIGNIFICANCE AND NOVELTY
As it stands, the results are novel and important enough for this journal.
